# Molecularly Imprinted Ligand-Free Nanogels for Recognizing Bee Venom-Originated Phospholipase A2 Enzyme

**DOI:** 10.3390/polym14194200

**Published:** 2022-10-07

**Authors:** Anamaria Zaharia, Ana-Mihaela Gavrila, Iuliana Caras, Bogdan Trica, Anita-Laura Chiriac, Catalina Ioana Gifu, Iulia Elena Neblea, Elena-Bianca Stoica, Sorin Viorel Dolana, Tanta-Verona Iordache

**Affiliations:** 1National Institute for Research & Development in Chemistry and Petrochemistry ICECHIM, Advanced Polymer Materials and Polymer Recycling Group, Spl. Independentei 202, 6th District, 060021 Bucharest, Romania; 2National Institute for Medico-Military Research and Development “Cantacuzino”, Spl. Independentei 103, 5th District, 011061 Bucharest, Romania

**Keywords:** molecularly imprinted polymers, ligand-free nanogels, bee venom phospholipase A2, synthetic antivenom, bee envenomation

## Abstract

In this study, ligand-free nanogels (LFNGs) as potential antivenom mimics were developed with the aim of preventing hypersensitivity and other side effects following massive bee attacks. For this purpose, poly (ethylene glycol) diacrylate was chosen as a main synthetic biocompatible matrix to prepare the experimental LFNGs. The overall concept uses inverse mini-emulsion polymerization as the main route to deliver nanogel caps with complementary cavities for phospholipase A2 (PLA2) from bee venom, created artificially with the use of molecular imprinting (MI) technologies. The morphology and the hydrodynamic features of the nanogels were confirmed by transmission electron microscopy (TEM) and dynamic light scattering (DLS) analysis. The following rebinding experiments evidenced the specificity of molecularly imprinted LFNG for PLA2, with rebinding capacities up to 8-fold higher compared to the reference non-imprinted nanogel, while the in vitro binding assays of PLA2 from commercial bee venom indicated that such synthetic nanogels are able to recognize and retain the targeted PLA2 enzyme. The results were finally collaborated with in vitro cell-viability experiments and resulted in a strong belief that such LFNG may actually be used for future therapies against bee envenomation.

## 1. Introduction

Hymenoptera (bee/wasp/ant) envenomation is not usually lethal for humans and animals if the venom intake is lower than the lethal dose [1]. However, it is well known that the venom from the Hymenoptera insects is a potent neurotoxin and that the main destructive component is the specific secreted phospholipase A2 (PLA2) [2]. Bee venom PLA2 enzyme acts synergistically with the polyvalent cations (toxins) in the venom [3], creating an increased hemolytic effect and quick access of toxins into the blood flow, targeting important organs such as the brain, kidney and liver [4]. This enzyme simply degrades the cellular phospholipidic membranes and in high amounts, as in envenomation, causes decreased blood pressure and thereafter inhibits blood coagulation [5,6]. Therefore, by removing a high amount of PLA2 enzyme from sting/bite zone, the rest of the venom toxins can be locally blocked, and since the phospholipidic membranes are stopped or retarded from degradation, the toxins and other allergens will have limited access into the blood flow. For this reason, the present study targets the development of complementary, or even alternative, antivenom therapies that can reduce the quantity of the toxic PLA2 enzyme intake before the phospholipidic membranes are damaged.

In spite of recent technological developments, no effective and safe therapies are currently available for treating the victims of mass honeybee or wasp attacks [7]. Adrenalin is the first-aid treatment of choice for the systemic allergic response with dyspnea and/or hypertension, while in patients without anaphylaxis, the suggested conservative approach is based on observation and treatment of symptoms [8]. Since 1996, multiple attempts to create antivenom as an emergency treatment for bee envenomation were proposed [9,10,11]. Antivenom is created by injection of sublethal toxin doses into an animal such as a sheep or horse, followed by harvesting the blood serum of the animal, which contains significant quantities of toxin-recognizing antibodies [12]. However, most of the studies regarding antivenom production suggested that the reason for their ineffectiveness is linked to the low immunogenicity of targeted bee venom toxins [13]. Recently, noteworthy antivenom designs based on monoclonal or oligoclonal antibodies [14,15] have emerged and may contribute to new and effective bee envenomation therapies. Yet, the technology is still young and needs serious efforts to deliver viable antivenom therapies [13]. Meanwhile, a great deal of research was focused on developing nano-sized hydrogels known in the literature under the name of nanogels [16,17]. The most common applications of nanogels, with controlled release properties of various active principles, are found in the tissue engineering field, biomedical implantology and bionanotechnology [16]. Having aside the newest trends in nanogels development [18,19,20] and access to technologies that allow for creating synthetic antibodies in various polymer networks, i.e., molecular imprinting (MI) [21,22], this work aims to prepare ligand-free nanogels (LFNGs) with complementary specific binding sites for PLA2, named molecularly imprinted polymers (MIPs). Synthetic nanogel antibodies make it possible to directionally modify the molecular size, affinity, specificity, and immunogenicity and effector functions of a natural antibody, as well as to combine antibodies with other functional agents for diagnosing and treating various diseases, particularly using new technologies meant to refine the effector functions of therapeutic antibodies [23]. The advantages of such synthetic antibodies include lower manufacturing costs, a medium level of synthesis complexity and no specific requirements for storage and transportation, as compared to the traditional antivenom. On the other hand, the proposed systems, denoted molecularly imprinted polymer ligand-free nanogels (MIP-LFNGs), are free of template molecules or ligands and can retain various compounds using the matrix itself for targeting [24,25].

Thereby, the novelty introduced by this pioneering work is the exploration of combined methods and concepts of state-of-the-art nanotechnologies and molecular imprinting techniques to deliver novel, efficient and cheaper antivenom variants for bee envenomation. The MIP-LFNG for bee venom-originated PLA2 recognition and retention, as presented in this study, is an original concept and has never been reported as a potential therapy against bee envenomation. To this day, only a few reports dealing with synthetic polymer nanoparticles as plastic antibodies with the capacity to bring and neutralize the hemolytic toxin melittin peptide were reported by Hoshino et al. [26,27,28,29]. In this respect, polymer nanoparticles (NPs) were prepared using N-isopropylacrylamide (NIPAm) as a core polymer combined with N-t-butylacrylamide, acrylic acid or N-3-aminopropyl methacrylamide as a functional monomer [26,27,29]. The resulting NPs, with an average size of 50 nm, were able to bind melittin in high amounts, up to 180 μmol/g NP [26]. In some early studies of this group [27,28], NPs based on NIPAm were molecularly imprinted with melittin and labeled with fluorescein *o*-acrylate to evaluate the binding of melittin and the behavior of the NPs in vivo. Other recent studies on molecularly imprinted nanogels were reported by Takeuchi’s group for protein recognition as well [30,31]. In these cases, the authors prepared MIP nanogels of about 45 nm diameter possessing good binding affinity and specificity (F > 20 at 1 mg/mL polymer, but low reaction yields below 1%) capable of protein corona regulation via albumin recognition. Nevertheless, the latter studies have successfully detailed some meaningful insights related to nanoparticle–cell interactions with the emphasis on the cellular uptake mechanism in cancer cells and immune-related cell lines [30], followed by in vivo studies revealing the uptake of albumin in MIP nanogels and their targeting ability for tumor tissue [31].

## 2. Materials and Methods

### 2.1. Materials

In order to obtain the MIP-LFNGs, polyethyleneglycol diacrylate (PEGDA, MW = 700 g/mol), sorbitan monooleate (SPAN 80), N,N,N′,N′-tetramethyl ethylenediamine (TMEDA) sodium chloride (>99%), cyclohexane (CHx, 99.5%), phospholipase A2 from bee venom Apis Mellifera (PLA2), 2-amino-2-(hydroxymethyl)-1,3-propanediol (TRIS, 99.8%), hydrochloric acid (HCl, 37%) and acetone (99,92%) were purchased from Sigma-Aldrich (St. Louis, MO, USA). Ammonium persulfate (APS, 98%) was purchased from Peking Chemical Works (Beijing Chemical Works, Beijing, China), polyethylene glycol sorbitan monooleate (TWEEN 80, oleic acid, ≥58.0%) was purchased from Sigma-Aldrich (St. Louis, MO, USA) and phosphate-buffered saline (PBS) was purchased from Roti-CEL (Karlsruhe, Germany). Invitrogen EnzChek Phospholipase A2 Assay Kit was purchased from Thermo Scientific LSG (Life Technologies Ltd., Inchinnan Business Park Paisley, UK). Bee venom (BV) was used in the form of lyophilized powder and was purchased from The Research and Development Institute for Beekeeping (Bucharest, Romania). The difunctional macromer polyethyleneglycol diacrylate (PEGDA, MW = 2000 g/mol) was synthesized as previously reported by Radu et al. [32] (results of molecular weight, functionality, structure and thermal stability are provided in the Appendix A).

### 2.2. Synthesis and Purification of LFNGs for Recognizing and Retaining Bee Venom-Originated PLA2

LFNGs were prepared similarly to the recipe previously described by our group, but with some modifications [33]. The organic phase was prepared by dissolution of the mixed emulsifiers SPAN80 (0.9 mol/L) and TWEEN80 (0.09 mol/L) in a 7:1 ratio (*w*/*w*) with cyclohexane (9 mol/L) in a round glass-bottom reactor. This mix was homogenized by magnetic stirring at 600 rpm, degassed and purged with nitrogen for 10 min. Meanwhile, the aqueous phase was prepared by dissolution the difunctional macromers PEGDA 700/PEGDA 2000 (0.4 mol/L), TMEDA (0.04 mol/L) and NaCl (0.2 mol/L) in ultrapure water (2 mL). After 10 min of mechanical stirring (600 rpm), degassing and purging with nitrogen gas, the aqueous phase was added to the organic phase under magnetic stirring (600 rpm). In order to prepare the molecularly imprinted polymer LFNGs (MIP-LFNGs), PLA2 (1:5 molar ratio PLA2:PEGDA) was prepared separately in ultrapure water (1 mL) or TRIS/HCl (pH 8.2; 50 mM), after which it was added to the reaction mixture under quick magnetic stirring (600 rpm). The rationale behind using water and the variant of a buffer solution to solubilize the PLA2 was to maintain as much as possible the conformation of PLA2 that may be found in the venom, in order to increase the specificity of the final nanogels for PLA2. The polymerization reaction was initiated by APS (5% w. relative to PEGDA), after which the stirring rate was lowered to 200 rpm and the temperature was set at 30 °C. The mini-emulsion was maintained in the previously mentioned conditions for 42 h and subsequently centrifuged. The supernatant was removed, and the nanogel phase was washed with cyclohexane, acetone and ultrapure water to remove the continuous media, the emulsifiers, the template PLA2 enzyme and any unreacted macromer. Thus, prepared MIP-LFNGs were lyophilized for 48 h to yield a powder. Furthermore, another nanogel system, called non-imprinted polymer LFNG (NIP-LFNG), was prepared as a reference; in this case, PLA2 was not added during the polymerization. The recipes for preparing the MIP-LFNG W or T (where W stands for water and T for TRIS/HCl solution) and for the reference NIP-LFNG are also summarized in Table 1.

### 2.3. Characterization Methods and Instruments

#### 2.3.1. Structural and Morphological Characterization of LFNGs

Fourier transform infrared (FT-IR) spectra, recorded on a Bruker Vertex 70 instrument in the 400–4000 cm^−1^ range with 4 cm^−1^ resolution and 32 scans (on KBr pellets), were useful for highlighting the molecular imprinting effect.

Thermogravimetric analyses (TGA) were carried out by using a Thermal Analysis SDT600 instrument and heating 5–10 mg samples from 30 °C to 1000 °C at a heating rate of 10 °C/min under nitrogen flow.

The particle sizes of LFNGs were determined by dynamic light scattering (DLS) analyses using a Malvern Zetasizer Nano-ZS system equipped with a 4 mW He-Ne laser (633 nm). All measurements were performed in five replicates, and the results are reported as the mean together with the standard deviation.

Transmission electron microscopy (TEM) pictures were taken using a Tecnai G2 F20 TWIN Cryo-TEM. Two protocols were used. The first one consisted of directly sampling the emulsion and placing it on a carbon-film-covered grid. The excess emulsifiers were removed by 5 s immersion of the grid in acetone. The second protocol consisted of the redispersion of the purified nanogels in distilled water and placement of the sample on the same type of grid.

#### 2.3.2. Batch Binding Experiments Assisted by Activity Measurements of PLA2

Binding experiments were performed in order to investigate the specificity and capacity of MIP-LFNG and NIP-LFNG to recognize and rebind PLA2. In this respect, the assays were based on measuring the decrease in activity in PLA2 aqueous solutions or bee venom solutions (U/mL) before and after contact with the nanogels, using the EnzChek Phospholipase A2 Assay Kit, at a plate dilution of 1/2 (U/mL) (or 1/40) initial solution. In brief, the binding experiments involved contacting 10 mg of each LFNG with 1 mL pure PLA2 or bee venom solution of known concentration (0.1 mg/mL PLA2 enzyme and 1 mg/mL bee venom). The supernatants were collected after 15 and 30 min, diluted (1/40) and analyzed by fluorescence for changes in the emission intensity ratio at 490 nm with excitation at 450 nm. To quantify the adsorbed PLA2 (also known as the rebinding capacity, Q (U PLA2/g nanogel)), for MIP-LFNGs and corresponding NIP-LFNG, the study presented the hypothesis that the decrease in activity as measured at 450 nm and 490 nm was due to the nanogels’ specific adsorption of PLA2. The method for calculating Q is given in Equation (1), where Ci (U/mL) represents the concentration of PLA2 in the reference aqueous solution or venom solution, Cf (U/mL) represents the concentration of PLA2 after contact with nanogels, mp (g) is the nanogel weight and Vs (L) is the volume of the feed solution (see also Appendix A).
(1)$$Q=\left(c_{i}-c_{f}\right)\cdot V_{S}/m_{p}$$

The imprinting factor, F, expressed by Equation (2), quantified the specificity with which MIP-LFNGs rebind PLA2, compared to the corresponding NIP-LFNG, where QMIP and QNIP are the rebinding capacities of MIP-LFNG (W or T) and NIP-LFNG, respectively. The binding experiments were carried out in duplicate, using fluorescence measurements on a reader for Tecan Infinite M1000 microplates. The results, as mean values of two replicates, were expressed as U/mL after extrapolation on a standard curve made with a standard solution of PLA2.
(2)$$F=Q_{MIP}/Q_{NIP}$$

#### 2.3.3. Cytotoxicity Study of LFNGs

Extracts from LFNGs were obtained by placing the materials in Dulbecco’s Modified Eagle Medium (DMEM), 5 mg/mL, for 24 h at 4 °C and collecting the supernatants by centrifugation (15 min, 10,000× *g*). Mouse fibroblast cell line L929 (ECACC 85011425) was used to test the cytotoxicity of LFNGs. L929 cells were cultured in DMEM supplemented with 10% fetal bovine serum (FBS), 100 μg/mL penicillin, 100 μg/mL streptomycin and 1 mM L-glutamine in a 5% CO_2_ atmosphere incubator at 37 °C. Cells were enzymatically detached and seeded in 96-well culture plates at a density of 1 × 104 cells/well and cultured overnight. Subsequently, the supernatant was discarded and replaced with binary dilutions (of LFNG extracts). After overnight incubation, a 3-(4,5-dimethylthiazol-2-yl)-2,5-diphenyltetrazolium bromide (MTT) assay was performed to evaluate the cell viability. Briefly, the supernatant was discarded and replaced with DMEM containing 0.5 mg/mL MTT. The assay is based on the ability of NAD(P)H-dependent oxidoreductase enzymes in living cells to reduce yellow tetrazolium salts to purple formazan crystals. Cells were incubated for an additional 3 h, and then lysis solution (20% *w*/*v* sodium dodecyl sulfate, 50% *w*/*v* N,N-dimethylformamide, 0.4% *w*/*v* acetic acid, 0.04 M hydrochloric acid) was added to dissolve the insoluble formazan crystals and the resulting-colored solution was quantified by measuring the absorbance at 570 nm using a microplate spectrophotometer (Multiskan FC, Thermo Scientific). The percentage of cell viability for each experimental condition was calculated by setting the control as 100%.

## 3. Results

### 3.1. Synthesis of LFNGs

In the current study, we aimed to combine the advantages of more efficient treatments based on nanomaterials and the specificity of MIPs for the development of molecularly imprinted ligand-free nanogels (MIP-LFNGs) for recognizing and retaining bee venom-originated PLA2. The inverse mini-emulsion polymerization system involved the formulation of a stable mixture, composed of droplets of polymer aqueous solution suspended by a mixture of co-surfactants in a continuous organic medium [33,34]. Herein, we investigated (i) the synthesis of LFNG based on poly (ethylene glycol) diacrylate initiated by a redox initiator system at body temperature, in the absence or presence of PLA2 enzyme (according to Figure 1); (ii) the morphology and structure of LFNGs; (iii) the performance of prepared LFNGs by single-enzyme rebinding experiments and by specific rebinding from bee venom; and the (iv) in vitro cytotoxic effect of LFNGs.

The following work describes the optimized recipes resulting from many variants of nanogel synthesis. In this respect, the same recipes were used to prepare nanogels using either PEDGA 700 or 2000 alone. However, the samples were discarded after performing DLS (Appendix A) and TEM analysis (Appendix A, Appendix A) which clearly showed that the systems were not proper for the studied application in terms of average particle size, polydispersity and uniformity.

### 3.2. Structural and Morphological Characterization of LFNGs

#### 3.2.1. FT-IR Spectroscopy

The LFNG series were evaluated by FT-IR spectroscopy (Figure 1). In the spectrum of the NIP-LFNG, the bands generated by the C-H stretching vibrations around 2870 cm^−1^, the carbonyl group (C=O) stretching vibration at 1729 cm^−1^ [35] and C-O stretching vibration at 1097 cm^−1^ can be clearly distinguished. The characteristic band of OH stretching vibrations from poly (ethylene glycol) was registered around 3500–3400 cm^−1^, while the bands assigned to -C=C- at 1620 cm^−1^ completely disappeared from the LFNG spectrum, indicating the consumption of -C=C- bonds of PEGDA during polymerization [36,37].

The FT-IR spectrum of the PLA2 template presented intense characteristic bands as well. The bands at 3289 and 3072 cm^−1^ correspond to the overlapping O–H stretching vibrations and the N–H stretching vibrations in amide A (more intense) and amide B of proteins. The intense band at 3289 cm^−1^ is the result of resonance between N–H stretching and the overtone of amide II [38,39]. The band at 2924 cm^−1^ is characteristic of symmetric (CH_2_) groups, while the bands at 1641 and 1533 cm^−1^ (characteristic of the peptide amines and amino acids) correspond to the C=O symmetric stretching vibrations of α-helical structure (amide I) and the N–H in-plane bending and C–N stretching of amino acids (amide II), respectively [40,41]

Interestingly, the spectra of both imprinted nanogels, i.e., MIP-LFNG (T) and MIP-LFNS (W), analyzed before the extraction of the PLA2 template, showed important changes compared to the reference nanogels, NIP-LFNGs, as well as to the same imprinted nanogels analyzed after PLA2 extraction, which indicated an efficient imprinting of PLA2. At a first glance, a broad band between 3000 and 3700 cm^−1^, similar to that observed in the spectrum of PLA2, was registered for the imprinted nanogels. This band was assigned to the overlapping bands of the O–H stretching vibrations and the N–H stretching vibrations in amide A and amide B of proteins. Furthermore, the characteristic band of -CH_2_ groups (at 2924 cm^−1^) and the one associated with the amide I bands (at 1641cm^−1^) were also spotted in the spectra of both imprinted nanogels before PLA2 extraction.

On the other hand, the FT-IR spectra of the imprinted nanogels analyzed after the extraction of the PLA2 template, named MIP-LFNG (T, ext) and MIP-LFNG (W, ext), are similar to those of the non-imprinted nanogel references, without any characteristic bands of PLA2. This resemblance proves the fact that the chemical structure of the imprinted nanogel matrix was not modified during the imprinting process (this process being non-covalent) and also that a proper extraction of PLA2 from the nanogels was performed with the aim of cleaving the specific binding sites thus created [42].

Thereby, it can be assumed that the imprinting of PLA2 was successful and that specific binding sites were created, considering that the prominent features of PLA2 are present in the spectra of imprinted nanogels before template extraction [43] and disappear after nanogels are thoroughly washed.

#### 3.2.2. TGA Investigation

The thermal stability of LFNGs was highlighted by TGA/DTG analysis provided in Figure 2a,b. The results of TGA and the corresponding derivative curves of NIP-LFNG showed a slight decrease in thermal stability, while the MIP-LFNG (T, ext) revealed a similar thermal stability to that of MIP-LFNG (T), before PLA2 extraction (Figure 2). The NIP-LFNGs presented one small shoulder at 163 °C (Tmax) attributed to polymer lose chain, after which it maintained an expected decomposition trend centered at 392 °C (Tmax) that can be related to the degradation of the polymer backbone chain, with a final weight loss of 95% [44,45,46].

The TGA and derivative curve of the PLA2 enzyme revealed the decomposition of the amino acids in several stages, as follows [47]: the first stage at 222 °C was attributed to the elimination of NH3 and the formation of unsaturated acids; this stage was followed by the release of intramolecular water and the formation of lactams, and ultimately the decarboxylation process occurred, resulting in the formation of amines at 307 °C.

MIP-LFNGs (T, ext) analyzed after the extraction of the PLA2 template exhibited a very small decomposition step in the vicinity of 160 °C followed by the main decomposition step at a maximum temperature of 392 °C (Tmax), with a final weight loss of 82%. The stability of the extracted nanogels, MIP-LFNGs (T, ext), was similar to that of the reference nanogels, NIP-LFNGs, which confirmed the FT-IR observations and conclusions referring to the fact that the chemical structure and composition of the imprinted nanogel matrix was not significantly modified during the imprinting process. 

Interestingly, the MIP-LFNG (T) before the extraction of the PLA2 template revealed a small shoulder at 248 °C (Tmax), which was attributed to the very low amounts of PLA2 used for the imprinting process, while the main decomposition process of the nanogel matrix followed a similar trend to that of the NIP-LFNGs, but with a lower maximum temperature for decomposition, at 382 °C (Tmax), and a higher final weight loss of around 89% compared to the extracted MIP-LFNGs (T, ext). The slight decrease in thermal stability of about 10 °C for the nanogels analyzed before PLA2 extraction may be due to the presence of the polymer–template interaction between the amino acid side chains of PLA2 specifically involved in the binding to the functional groups of the nanogel matrix [27,48,49]. 

#### 3.2.3. DLS Investigation

The particle size distribution and the polydispersity (PDI) of the LFNGs before and after purification were investigated using DLS, given the targeted application. As shown in Table 2, the synthesized LFNGs exhibited low PDI values, and no significant coagulation was observed during polymerization. The PDI of LFNGs should be very low (under 0.5, which means that the nanogels have similar sizes) in order to obtain comparable results in each batch, but also to decrease the potential cytotoxic effects of nanogels given by their uneven size, as demonstrated by other studies as well [33]. The Z-average particle size of the LFNG was registered within the desired range, below 200 nm (and 143 ÷ 198 nm, in this case), and the PDI was below 0.375. The Z-average particle size of the MIP-LFNG (T) and MIP-LFNG (W) before the PLA2 template extraction was approximately 189 nm and 198 nm, respectively, yet slightly bigger than that of NIP-LFNGs. This observation can be linked to the presence of the PLA2 template in the structure of the synthesized LFNGs, which also contributed to the increase in the hydrodynamic volume of the analyzed nanogels. The latter hypothesis was also sustained by the fact that for the MIP-LFNGs analyzed after PLA2 extraction, namely MIP-LFNG (T, ext) and MIP-LFNG (W, ext), the Z-average particle size decreased in the size range of 163–170 nm. It may also be noted that the 20–30 nm difference between the average sizes of extracted MIP-LFNGs and the average size registered for the reference NIP-LFNG may be due to the hydrodynamic volume occupied by the cleaved imprinted cavities specific for PLA2.

#### 3.2.4. TEM Images

Transmission electron microscopy (TEM) images of nanogels analyzed directly from emulsion and after purification also supported the previously discussed results regarding the average size of nanogels and the imprinting process (Figure 3). The micrographs of the NIP-LFNG taken directly from the final emulsion (Figure 3a,b) revealed the presence of individual spherical nanogels, having dimensions roughly in the range of 60–180 nm. It is important to note the presence of emulsifiers as needle-shaped formations, which form a continuous layer around the synthesized nanogels. After washing, which implied the removal of emulsifiers as well, only the spherical nanogels could be distinguished; they had no significant morphology modifications but were slightly agglomerated and had similar dimensions (Figure 3c). 

Meanwhile, the micrographs of the MIP-LFNG (W) and MIP-LFNG (T) before and after the PLA2 extraction showed significant morphological changes as compared to the samples analyzed directly from emulsion (Figure 3d–i). Both types of MIP-LFNGs indicate a spherical shape morphology, having dimensions roughly in the range of 90–190 nm. In the case of the LFNGs taken directly from the final emulsion (Figure 3d,e,g,h), the presence of needle-shaped emulsifiers takes an interesting microstructural arrangement in tube form, which can also be due to the presence of the PLA2 enzyme. Thus, on the surface of LFNG spheres, the microcrystals with denser and more homogeneous structures may actually be PLA2 molecules frozen in their crystalline state [50]. These micrographs, probably the first of their kind, show how the PLA2 enzyme binds to the nanogel matrix and, subsequently, leaves marks of its interaction by creating molecularly imprinted cavities (Figure 3f,i). TEM micrographs of the MIP-LFNGs after the PLA2 extraction also showed the presence of multiple spherical zones with different electron densities that sustain the latter affirmation [51]. Therefore, this structural detail is proof of the non-covalent interactions between the template and macromers and can be suggestive and characteristic of the presence of imprinted free nanocavities on the surface of synthesized MIP-LFNGs [52].

### 3.3. Binding Properties of LFNGs

The binding properties of PLA2 investigated in an aqueous medium in batch mode were determined for the two types of MIP-LFNGs (i.e., MIP-LFNG (T) and MIP-LFNG (W)) and the corresponding blank NIP-LFNG. The specificity for PLA2 uptake was assessed by quantifying the activity of PLA2 in solutions before and after contact with the nanogels (as presented in Appendix A). Therefore, the adsorption capacity, Q (U/g^−^), and imprinting factor, F, of nanogels were also given as units of PLA2 (Figure 4a,b).

Both imprinted nanogels, MIP-LFNG (T) and MIP-LFNG (W), exhibited higher affinity for PLA2 when contacted with an aqueous PLA2 solution than the corresponding NIP-LFNG, leading to impressive rebinding capacities after 30 min of exposure of 39.93 U/mg (490 nm) and 38.66 U/mg (450) for MIP-LFNG (T) and 39.49 U/mg (490 nm) and 38.36 U/mg (450) for MIP-LFNG (W). Due to the low PLA2 amounts adsorbed by the NIP-LFNG after 30 min of exposure, i.e., 5.00 U/mg (490 nm) and 4.52 U/mg (450 nm), the imprinting factor, F, values calculated for MIP-LFNG (T) and MIP-LFNG (W) after 30 min of exposure were close to 8 (Figure 4a,b), meaning that MIP-LFNGs recognize and retain PLA2 about 8-fold more specifically than the reference NIP-LFNG. The resulting values are comparable to the results of other authors related to molecularly imprinted nanogels for peptide recognition [30,31]. It is also important to mention that the two investigated parameters continue to improve with time; an increasing trend was observed from 15 min exposure time to 30 min, indicating that an adsorption equilibrium was surely attained after 30 min of exposure.

The following in vitro experiments of PLA2 binding from bee venom have provided important information regarding the potential of such MIP-LFNGs to retain specifically the enzyme directly from the venom. Although the values of activity measured after contact with the venom (Appendix A) were not as spectacular as the ones recorded for the binding assays from PLA2 solutions, a similar trend was observed with regard to the performance of each nanogel system. Figure 5a,b presents the decrease in PLA2 activity (%) for each nanogel system, after exposure to venom at 15 and 30 min, relative to the initial PLA2 activity in the bee venom (100%). However, it was somewhat surprising that the specificity of MIP-LFNGs (W) for PLA2 retention has dropped significantly as compared to the previous assay; their retention capacity for PLA2 this time is close to the retention capacity of NIP-LFNG (see the activity drop of PLA2 after exposure to venom in Figure 5a,b).

### 3.4. Cytotoxicity Assay

The potential cytotoxicity of LFNGs was also studied as a result of the targeted application, i.e., as alternatives to traditional antivenom which is administrated intravenously. In this respect, the effect of LFNG concentrations on cell viability was investigated by MTT assay (Figure 6). After 24 h of incubation at different dilutions (1/4, 1/8, 1/16 and 1/32), only a slight reduction in the cell viability was observed, less than 3% (values given in Appendix A, Appendix A). The results were, however, significant because the L929 cells remained with high viability (≥97%) as shown in Figure 6, even at high nanogel concentrations of 1/4, in which case a 98.29 ± 1.33% and 100.8 ± 1.3% cell viability was registered for MIP-LFNG (T) and MIP-LFNG (W), respectively. Other studies reported similar results when using PEGDA-based nanoparticles [30,53]. A very slight and odd decrease in cell viability was observed at higher dilutions for MIP-LFNG (W), i.e., down to 97.6 ± 1.7% at 1/16 but, still, very close to the reference. What is also interesting to note is the fact that LFNGs, particularly NIP-LFNG and MIP-LFNG (T), led to an increase in cell viability, especially at higher nanoparticle dilutions, which means that the two systems were also able to induce slight cell proliferation (no more than 9%). Yet, this property may help in the administration of the synthetic antivenom and be of benefit in a secondary activity of cell restoration/proliferation after PLA2 damage to existing viable cells.

## 4. Conclusions

In conclusion, this study reports the development of original ligand-free nanogel systems molecularly imprinted with bee venom-originated PLA2 (MIP-LFNGs) as a potential therapy for bee envenomation. In this respect, the nanogels were prepared by a known technique (mini-emulsion polymerization) that can deliver spherical nanosized gels with narrow polydispersity, while the polymer matrix consisted of a mixture between two macromers of PEGDA with two different molecular weights, i.e., 700 and 2000 g/mol. Thus, the nanogels prepared in the presence or absence of the PLA2 template, called MIP-LFNGs and NIP-LFNGs, respectively, were analyzed in terms of structure, composition, morphology and particle size in order to gain a better understanding of their behavior when submitted to rebinding assays of PLA2 from aqueous solution or bee venom and to cytotoxicity investigations. FT-IR, TGA, DLS and TEM analysis have pointed out that specific imprinted cavities for PLA2 retention were created in both nanogel systems, i.e., MIP3-LFNG (T) and MIP-LFNG (W). However, the system denoted MIP-LFNG (T), developed using the PLA2 template solubilized in TRIS/HCl, seems to perform better during the rebinding assays, retaining PLA2 from solution 8.5-fold more specifically than the non-imprinted reference, NIP-LFNG, and attaining a high rebinding capacity of approximately 40 U PLA2/mg of nanogel. The differences between this system and the one denoted MIP-LFNG (W), developed using the PLA2 template solubilized in water, were very small with the exception of the capacity of rebinding the PLA2 from venom, in which case MIP-LFNG (T) reduced the activity of PLA2 in the bee venom by almost 10% (compared to the 3% registered for MIP-LFNG (W)); thus, MIP-LFNG (T) was about 3 times more efficient than MIP-LFNG (W) in recognizing and retaining the PLA2 from the venom. Furthermore, the cytotoxicity of MIP-LFNGs was very low compared to the reference, even at high nanogel concentrations, whereas the lowest values for cell viability were registered for MIP-LFNG (W) at a dilution of 1/16 (97.6 ± 1.7%).

## Data Availability

The data presented in this study are available on request from the corresponding author.

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
