# Peer review of "Molecularly Imprinted Ligand-Free Nanogels for Recognizing Bee Venom-Originated Phospholipase A2 Enzyme"

_polymers, 2022, doi:10.3390/polym14194200_

Round 1
Reviewer 1 Report
I have no additional comments.
Author Response
Thank you for your recommendations. The manuscript was checked for English spelling and some other mistakes. The modifications were highlighter in red (with Track Changes).

Reviewer 2 Report
Comments and Suggestions for Authors
The paper entitled ‘Molecularly Imprinted Ligand - Free Nanogels for recognizing bee venom-originated phospholipase A2 enzyme’ developed a original ligand-free nanogel systems molecularly imprinted with bee-venom originated PLA2 (MIP-LFNGs) to serve as potential therapy for bee envenomation in drug delivery field. The manuscript focuses on an interesting and current topic in the era of bee venom. Following substantial revisions should be incorporated in the manuscript prior to acceptance.
(1) This work in interesting. I have no specific concern but it is mandatory to correct the manuscript in this point: Please review other methods in bee envenomation delivery, MIPs should not be the first method, the back ground should be added in introduction section.
(2) LINE 37-39 Please provide the citation.
(3) LINE 72-74 Please provide more infomation on these reports.
(4) LINE 74-75 Please provide more infomation on these reports.
Author Response
Thank you for your recommendations.
The modifications were highlighter in red (with Track Changes).
1. This work in interesting. I have no specific concern but it is mandatory to correct the manuscript in this point:Please review other methods in bee envenomation delivery, MIPs should not be the first method, the back ground should be added in introduction section.
"In spite of recent technological developments, no effective and safe therapies are currently available for treating the victims of mass honeybee or wasp attacks [7] 7. Almeida RAM de B, Olivo TET, Mendes RP, Barraviera SRCS, Souza L do R, Martins JG, et al. Africanized honeybee stings: how to treat them. Rev Soc Bras Med Trop. (2011) 44:755–61. doi: 10.1590/S0037-86822011000600020. Adrenalin is the first-aid treatment of choice for the systemic allergic response with dyspnoea and/or hypertension, while in patients without anaphylaxis, the suggested conservative approach is based on observation and treatment of symptoms [8] 8. Muraro A, Roberts G, Worm M, Bilò MB, Brockow K, Rivas MF, et al. Anaphylaxis: guidelines from the European Academy of Allergy and Clinical Immunology. Allergy. (2014) 69:1026–45. doi: 10.1111/all.12437. Since 1996, multiple attempts on creating antivenom as emergency treatment for bee-envenomation were proposed [9-11]. 9. M.J. Schumacher, N.B. Egen, D. Tanner, “Neutralization of bee venom lethality by immune serum antibodies”, 1996, Am J Trop Med Hyg, 55: 197-201.; 10. Santos KS, Stephano MA, Marcelino JR, Ferreira VMR, Rocha T, Caricati C, et al. Production of the first effective hyperimmune equine serum antivenom against Africanized bees. PLoS ONE. (2013) 8:e79971. doi: 10.1371/journal.pone.0079971; 11. Laustsen AH, Gutiérrez JM, Knudsen C, Johansen KH, Bermúdez-Méndez E, Cerni FA, et al. Pros and cons of different therapeutic antibody formats for recombinant antivenom development. Toxicon. (2018) 146:151–75. doi: 10.1016/j.toxicon.2018.03.004. Antivenom is created by injection of sub-lethal toxin doses into an animal such as a sheep or horse, followed by harvesting the blood serum of the animal, which contains significant quantities of toxin-recognizing antibodies [12]. 12. Jones, R.G.A.; Corteling, R.L.; To, H.P.; Bhogal, G.; Landon, J. A novel Fab-based antivenom for the treatment of mass bee attacks, Am. J. Trop. Med. Hyg. 1999, 61, 361-366. https://doi.org/10.4269/ajtmh.1999.61.361. However, most of the studies regarding antivenom production suggested that the rea-son for their ineffectiveness is linked to the low immunogenicity of targeted bee venom toxins [13]. 13. Manuela B. Pucca, Felipe A. Cerni, Isadora S. Oliveira, Timothy P. Jenkins, Lídia Argemí, Christoffer V. Sørensen, Shirin Ahmadi, José E. Barbosa and Andreas H. Laustsen, Bee Updated: Current Knowledge on Bee Venom and Bee Envenoming Therapy, Front. Immunol., 2019, Sec. Vaccines and Molecular Therapeutics, https://doi.org/10.3389/fimmu.2019.02090 Recently, noteworthy antivenom designs based on monoclonal or oligoclonal antibodies [14, 15] 14. Pessenda G, Silva LC, Campos LB, Pacello EM, Pucca MB, Martinez EZ, et al. Human scFv antibodies (Afribumabs) against Africanized bee venom: advances in melittin recognition. Toxicon. (2016) 112:59–67. doi: 10.1016/j.toxicon.2016.01.062; 15. Jenkins T, Fryer T, Dehli R, Jürgensen J, Fuglsang-Madsen A, Føns S, et al. Toxin neutralization using alternative binding proteins. Toxins. (2019) 11:53. doi: 10.3390/toxins11010053 have emerged and may contribute to new and effective bee envenoming therapies. Yet, the technology is still young and needs serious efforts to deliver viable antivenom therapies [13]."
2. LINE 37-39 Please provide the citation.
The authors agree with the suggestion and, thus, provided the following references:
- Vanessa O. Zambelli, Gisele Picolo, Carlos A. H. Fernandes, Marcos R. M. Fontes and Yara Cury, Secreted Phospholipases A2 from Animal Venoms in Pain and Analgesia, Toxins (Basel). 2017 Dec; 9(12): 406. doi: 10.3390/toxins9120406
- Mingarro I, Pérez-Payá E, Pinilla C, Appel JR, Houghten RA, Blondelle SE. Activation of bee venom phospholipase A2 through a peptide-enzyme complex. FEBS Lett. (1995) 372:131–4. doi: 10.1016/0014-5793(95)00964-B
3. LINE 72-74 Please provide more infomation on these reports.
The reports were extended as recommended.
"To this day, only few reports dealing with synthetic polymer nanoparticles as plastic antibodies with the capacity to bring and neutralize the hemolytic toxin melittin peptide were reported by Hoshino and co. [26-29]. In this respect, polymer nanoparticles (NPs) were prepared using N-isopropylacrylamide (NIPAm) as core polymer combined with N-t-butylacrylamide, acrylic acid or N-3-aminopropyl methacrylamide as functional monomers [26, 27, 29]. The resulted NPs, with an average size of 50 nm were able to bind melittin in high amounts, up to 180 μmol/ g NP [26]. In some early studies of this group [27, 28], the NPs based on NIPAm were molecularly imprinted with melittin and labeled with fluorescein o-acrylate to evaluate the binding of melittin and behavior of NPs in vivo."
(4) LINE 74-75 Please provide more infomation on these reports.
"Other recent studies on molecularly imprinted nanogels were reported by Takeuchi’s group for protein recognition, as well [30,31]. In these cases, the authors prepared MIP nanogels of about 45 nm diameter possessing good binding affinity and specificity (F > 20 at 1 mg/ mL polymer, but low reaction yields below 1%) capable of protein corona regulation via albumin recognition. Nevertheless, the latter studies have successfully detailed some meaningful insights related to nanoparticle cell-interactions with the emphases on the cellular uptake mechanism on cancer cells and immune-related cell lines [30], followed by in vivo studies revealing the up-take of albumin in MIP nanogels and their targeting ability for tumor tissue [31]."

Reviewer 3 Report
Comments to the Authors
In this study, authors have presented ligand-free nanogels (LFNGs) as potential antivenom mimics were devel- 13 oped to prevent hypersensitivity and other side-effects following massive bee at- 14 tacks.
The manuscript is fascinating and has the potential to attract a broad audience. I recommend acceptance of the manuscript subjected to minor corrections.
Authors should thoroughly follow the proper numbering order; after section 3, section 5 conclusion is presented. Section 2.1 is italic, whereas section 2.3 is bold with no italic. Please check the entire manuscript for more errors. Both the tables shown in the manuscript have been assigned the label Table 1.
The technical portion of the manuscript is solid and complete. Thus a minor revision is only required.

Author Response
Authors should thoroughly follow the proper numbering order; after section 3, section 5 conclusion is presented. Section 2.1 is italic, whereas section 2.3 is bold with no italic. Please check the entire manuscript for more errors. Both the tables shown in the manuscript have been assigned the label Table 1.
Thank you for your observations. The authors have corrected all the errors and checked the entire manuscript for other mistakes. The modifications were highlighter in red (with Track Changes).

Reviewer 4 Report
In this study, ligand-free nanogels (LFNGs) as potential antivenom mimics were developed with the aim of preventing hypersensitivity and other side effects following massive bee attacks. The results were finally collaborated
with in vitro cell viability experiments and resulted in a strong belief that such LFNG may actually be used for future therapies against bee envenomation.
Comments:
what is the purity of acetone?
which polyethylene glycol sorbitan monooleate (TWEEN 80) did you use?...specify
Author Response
Thank you for your observations. The queries were answered below, and the modifications were highlighter in red in the manuscript (with Track Changes).
1. what is the purity of acetone?
The authors agree with the recommendation and, thus, inserted in Section 2.1. Materials the purity of acetone (99,92%)
2. which polyethylene glycol sorbitan monooleate (TWEEN 80) did you use?...specify
The TWEEN 80 product is the commercial product with oleic acid, ≥58.0%, which was originally purchased from Fluka, but now the parent company is Sigma-Aldrich. Since the product can only be now delivered by Sigma-Aldrich, we changed the distributor company to Sigma-Aldrich and added the content of oleic acid in brackets.
